# UNCERTAINTY-AWARE LLM PROBING

## ABSTRACT

Probing the hidden states of LLMs has been broadly studied in the literature. It is known to be an efficient and effective method to elicit linguistic knowledge and even higher-level behaviors from the models; for example, factual accuracy and uncertainty. However, most evaluations focus on specific short-form question answering scenarios. Further, the applicability of probes has been recently questioned, particularly in out-of-distribution (OOD) scenarios. Given their benefits, efficiency, and the variety of use cases, we study whether and when common uncertainty estimation techniques make probes more reliable. Moreover, we design a novel, robust gradient-based score, tailored to binary classification. We study various established uncertainty estimation techniques with a focus on probes for LLM correctness. Our evaluation covers many different LLMs and a variety of task domains with smaller distribution shifts and OOD scenarios. Most importantly, we find that, in LLM classifiers such as guardian models (i.e., a type of LLM judge to evaluate prompt safety), uncertainty estimators may considerably improve probe applicability. Generally, they however only work in domain. We specifically find that none of the methods studied works reliably under smaller distribution shifts. Overall, our score consistently ranks among the best performers.

## 1 INTRODUCTION

**LLM Probing.** It is widely understood that language models (LMs) do not necessarily communicate all their knowledge in their output. Similar to human gestures or facial expressions, model artifacts may provide additional information. For that reason, LMs have been probed, for example, to elicit knowledge present in their hidden states (Hewitt & Liang, 2019; Belinkov, 2022). Earlier works focused on linguistic knowledge and model understanding more generally (e.g., to check if models recognize erroneous source code (Troshin & Chirkova, 2022)). With large language models (LLMs) and their increasing utilization in high-stakes scenarios, model reliability and trustworthiness have become critical, and probing is thus increasingly studied by the AI community (Azaria & Mitchell, 2023; Gottesman & Geva, 2024). Most works train simple MLP probes to directly predict model correctness or – very related – uncertainty (Kossen et al., 2024). These predictions may serve as proxies for accuracy when labels are not available, with the latter being a common scenario.

However, recent evaluations have shown that the impressive probe performance in in-domain scenarios does not translate to out-of-distribution settings (Azizian et al., 2025; Heindrich et al., 2025; Orgad et al., 2025; CH-Wang et al., 2024). More specifically, it can degrade considerably with dataset shifts, even if the task the LLM is asked to perform stays generally the same (e.g., question answering). Azizian et al. (2025) provide an explanation by demonstrating that the directions learnt by linear probes for different datasets are often orthogonal to each other.

**Uncertainty-Aware Probes.** The potential of probes in LLMs motivates us to study their reliability and robustness more generally. In this paper, we consider uncertainty quantification (UQ) *on the probe*. While uncertainty quantification in deep neural networks has been studied in the past (Ovadia et al., 2019; He et al., 2025), probing is special. Hidden states are high-dimensional, compressed, task-entangled representations, which are very different from the dedicated features used in traditional classifiers. Further, hidden states may vary drastically across layers, prompt contexts, or domains. Hence, generalization is central, and it is specifically open, if the results from Ovadia et al. (2019), who study uncertainty quantification for classifiers under distribution shift (e.g., data perturbation) and out of distribution, translate to LLM probes. They show best results for ensembles. Finally, recent works observe that dedicated evaluations on the target modalities and in realistic scenarios

are needed (Malinin et al., 2022; Gustafsson et al., 2023; Seligmann et al., 2023). To the best of our knowledge, such insights about uncertainty quantification for probes are however rare. Wang et al. (2023) design Gaussian Process probes and evaluate them on image data, and Azizian et al. (2025) test conformal prediction on LLM probes. The latter turns out to be sub-optimal because conformal prediction assumes the scores to be ranked consistently across datasets; but it is not known when or why, or if alternative methods would be useful.

**Our Focus.** Given the diverse nature of LLM embeddings as described above, we hypothesize that we obtain more consistent results for probes when the underlying LLM is used as a classifier. That is, we then expect the hidden states to be more discriminative. Observe that there are many interesting and practically relevant such scenarios, we consider guardian models (i.e., a type of LLM judge to evaluate prompt safety) and also experiment with LLM judge scenarios. The tasks and prompt templates such LLMs are given tend to be fixed and constrained, making out-of-distribution scenarios more clearly defined and easier to analyze. Nevertheless, we also consider classical question-answering and chain-of-thought (CoT) reasoning over a wide set of tasks from LM Evaluation Harness (Gao et al., 2024). In terms of probe targets, we focus on LLM correctness; we also experiment with uncertainty and, in the context of the guardian, safety.

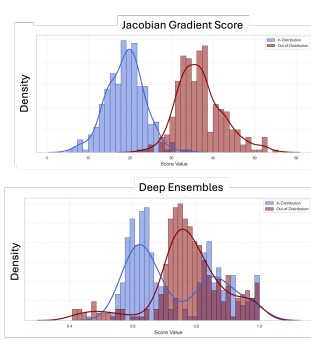

Figure 1: Our JDGrad score (top) separates ID and OOD data considerably better than deep ensembles (bottom).

We consider various types of models and data to get a broad overview of performance insights in various domains. Based on evaluations on uncertainty quantification in traditional ML, we would expect the uncertainty methods to work reliably in domain. Since LLMs are applied in the open world and probes have been shown to struggle out of distribution, we also consider distribution shifts and OOD scenarios. Note that we cannot expect the uncertainty scores to provide high-quality signals about probe (in)correctness in such settings, instead we should see generally high uncertainty.

**Our Contributions.** We study the applicability of LLM probes by evaluating and analyzing uncertainty-quantification methods in a variety of scenarios.

- We propose to also focus on LLMs as classifiers, such as guardian models for detecting harmful or ungrounded generations and LLM judges, since probing and uncertainty quantification are especially needed in safety-critical scenarios.
- We consider more general use cases by evaluating established UQ scores on a variety of LLMs (8b to 30b parameters) in common question-answering scenarios, mathematical problem solving, and chain-of-thought reasoning. Overall, we show that most UQ methods work well in domain.
- Moreover, we design a novel gradient-based score, tailored to binary classification. It ranks among the best, performs robustly, and provides useful OOD signals (Fig. 1). Note that existing gradient-based scores also provide such signals in our study, but they are not competitive in terms of UQ in domain. We further show an interesting connection between ADGrad and established margin-based classifiers.
- In a detailed case study, we describe how our scores provide meaningful signals in the challenging guardian tasks. More precisely, they help us detect ambiguous samples and label noise. scenarios,
- We identify one main problem with all the scores studied: they work in domain and some give OOD signals, yet we do not obtain reliable signals within domain under (slighter) distribution shifts in cases when the probes fail.
- Finally, we demonstrate how the probes' UQ scores can benefit probe applications.

## 2 PRELIMINARIES & RELATED WORK

**Notation.** Let $h \in \mathbb{R}^d$ denote hidden representations extracted from an LLM. Given an LLM $\mathcal{M} : \mathcal{X} \to \mathbb{R}^d$ mapping inputs to $d$-dimensional hidden representations, a probe $f_\theta : \mathbb{R}^d \to [0, 1]$ is a parametric function trained to predict task-specific properties from frozen representations. Formally, for hidden state $h = \mathcal{M}(x)$ and binary target $y \in \{0, 1\}$, the probe minimizes: $\theta^* = \arg\min_\theta \mathbb{E}_{(x,y)\sim\mathcal{D}}[\mathcal{L}(f_\theta(\mathcal{M}(x)), y)]$ where $\mathcal{L}$ is the binary cross-entropy loss and $\mathcal{D}$ the task distribution.

We seek uncertainty scores $s : \mathbb{R}^d \to \mathbb{R}_+$ that correlate with probe errors, enabling reliable deployment without ground-truth labels. An uncertainty score maps hidden states to scalar uncertainty estimates. We desire scores satisfying: $\mathbb{P}[\hat{y} \neq y | s(h) > \tau] > \mathbb{P}[\hat{y} \neq y | s(h) \leq \tau]$ for threshold $\tau$, where $\hat{y} = \mathbb{1}[f_\theta(h) > 0.5]$ is the predicted label. Given uncertainty scores, we construct a selective classifier: $g(h) = \hat{y}$ if $s(h) \leq \tau$, and $\perp$ otherwise; $\perp$ denotes abstention.

**Established Uncertainty Estimators.** We consider the below methods in our evaluation:

- **Deep Ensembles** Lakshminarayanan et al. (2017) train multiple independently initialized probes and aggregate predictions via averaging. Deep ensembles remain a simple and effective method for uncertainty estimation.

- **Conformal Prediction (CP)** Vovk et al. (2005); Papadopoulos et al. (2007) calibrates prediction sets using a held-out set, computing thresholds to guarantee coverage at level $1 - \alpha$ under exchangeability assumptions Bernardo (1996) – a weaker form of the i.i.d. assumption. We use the standard split conformal approach with logit scores from the probe's inference.

- **Evidential Deep Learning (EDL)** Sensoy et al. (2018); Amini et al. (2020), outputs evidence values for each class via a softplus activation, deriving Dirichlet parameters $\alpha = \text{evidence} + 1$. Epistemic uncertainty is computed as $K / \sum \alpha$ where $K$ is the number of classes, with KL regularization annealed during training. We specifically employ EDL because it has been shown that its training focuses on OOD detection Shen et al. (2024).

- **Local Ensembles** Madras et al. (2019) approximate the prediction variance of a full ensemble by exploiting local Hessian information by projecting the prediction gradient onto its low-curvature subspace. In practice, the method computes the top-$m$ eigenvectors of the Hessian (largest eigenvalues) using Lanczos iteration, and defines the ensemble subspace as their orthogonal complement, thereby efficiently capturing the flat directions without computing the full eigen decomposition.

- **Gaussian Process Probes (GPPs)** Wang et al. (2023) are classification probes tailored to include UQ. They employ Beta GPs with cosine kernels, fitting separate latent functions $f_\alpha, f_\beta$ for positive and negative classes. Uncertainty derives from the posterior variance after moment-matching.

Prior evaluations of UQ generalization for ML more generally have primarily focused on the established methods described above. In particular, ensembles have been shown to generalize best in the comparison of Ovadia et al. (2019) and proposed as most important baseline in the Shifts benchmark Malinin et al. (2022). Seligmann et al. (2023) focus on Bayesian methods, yet observe generally best results by adding ensembles on top of Bayesian approaches. In our analysis, ensembles similarly rank among the best metrics.

**Gradient-based UQ.** We extend prior work that computes gradient-based scores, which are simple to calculate post-hoc and have been shown to provide OOD signals. Lee & AlRegib (2020) consider confounding unseen labels and compute the squared L2 norm of the gradient as uncertainty score. Another basic approach, which we compare to in our experiments, is **GradNorm** (Huang et al., 2021). It is represented by the L1 norm of the gradients from the final layer, where the gradient is computed w.r.t. KL divergence between predictions and a uniform distribution. Follow-up papers introduced slight modifications. Igoe et al. (2022) suggest to scale the gradients of each class based on the corresponding predicted probabilities. Wang & Ji (2024) additionally consider different neural network layers and input perturbations to smooth the gradients. Moreover, they give theoretical guarantees by showing that (under certain assumptions) perturbation-based methods can approximate Bayesian neural networks, and that *the epistemic uncertainty derived by the expected gradient norm can serve as an upper bound compared to the uncertainty produced by perturbation-based methods* when the perturbations are small (see Prop. 3.5 in that paper). Most recently, Behpour et al. (2023) further focus on OOD detection and argue that the consideration of the full gradient space introduces unnecessary noise. They propose **GradOrth**, where they employ orthogonal gradient projections in the low-rank subspaces of ID data. For computing the gradient, they continue to use the uniform distribution already applied in GradNorm. Given the promising OOD detection capabilities, we continue this line of research in the context of LLM probes. In particular, we show that we can replace the KL term and also bypass the noise by suitably exploiting the binary classification setting.

**A Note on Uncertainty Estimation in LLMs.** As pointed out in the introduction, probes themselves are sometimes used for uncertainty prediction (Kossen et al., 2024) and, similarly, their correctness

predictions can be treated as uncertainty estimates for the LLM predictions (Xiong et al., 2024). This is different from our uncertainty estimates for the probe, but we take uncertainty estimation for the LLM as a use case in the evaluation (see Fig. 9): To show the effectiveness of the UQ on the correctness probe, we treat the latter as uncertainty estimate for the LLM prediction and compare it to existing LLM UQ scores.

# 3 GRADIENT-BASED UNCERTAINTY QUANTIFICATION FOR LLM PROBES

Based on the promising OOD detection capabilities of gradient-based UQ, we aim to tailor the idea to the probing setting. We design a simple and intuitive score, exploiting the binary classification setting.

Our gradient-based uncertainty scores similarly leverage the sensitivity of probe predictions to weight perturbations. For a probe $f_\theta : \mathbb{R}^d \to [0,1]$ with parameters $\theta$, we compute the Jacobian of the loss with respect to model weights, providing a direct measure of prediction stability. Given input embedding $h$ and a potential label $y \in \{0,1\}$, we compute a Frobenius norm as follows:

$$J_y(h) = \left\| \frac{\partial \mathcal{L}(f_\theta(h), y)}{\partial \theta} \right\|_F$$

where $\mathcal{L}$ is binary cross-entropy.

**JDGrad** ("Jacobian Difference"). Our uncertainty scores aggregate sensitivities across both possible labels as follows:

$$s_{\text{JDGrad}}(h) = |J_1(h) - J_0(h)|$$

Note that the binary classification setting allows us to drop the uniform or weighted distributions considered in related works (e.g., a good idea about alternatives and how to approach them is given in (Igoe et al., 2022)). Moreover, the difference has a direct interpretation. It captures how differently the model would need to adapt for opposing predictions, with high values indicating uncertainty. Intuitively, if $J_1(h) \gg J_0(h)$, the model is confident in predicting 0 (would require large updates to predict 1). When $J_1(h) \approx J_0(h)$, the model is uncertain.

Observe that the absolute values of this score offer faithful interpretations and we will show that they allow for OOD detection. However, the score is likely impacted by noise which stems from considering the full gradient space; recall that Behpour et al. (2023) therefore considered low-rank projections. We can indeed consider SVD and projections to obtain a more robust JDGrad score. However, in the following, we propose a more elegant solution for a second UQ score.

**ADGrad** ("Asymmetric Difference"). To normalize for varying gradient magnitudes across samples, we introduce:

$$s_{\text{ADGrad}}(h) = \frac{|J_1(h) - J_0(h)|}{J_1(h) + J_0(h) + \epsilon} = \frac{s_{\text{JDGrad}}}{J_1(h) + J_0(h) + \epsilon}$$

where $\epsilon$ prevents division by zero. This ratio captures the relative imbalance in gradient flows, being robust to absolute gradient scales.

**Robustness of ADGrad.** The primary advantage of ADGrad lies in its robustness and theoretical elegance. By normalizing the gradient difference, the score becomes invariant to the absolute magnitude of the gradients, which can vary significantly across different samples and model architectures, thus enabling more stable comparisons. More profoundly, we establish a theoretical connection to established margin-based metrics; see Appendix B. Under the single-sample regime (i.e., batch size $= 1$) and with the network in evaluation mode (i.e., batch normalization uses fixed statistics and no gradient accumulation occurs across samples), we prove that $s_{\text{ADGrad-bs1}}(h)$ exhibits layer invariance: for any given input $h$, the score remains constant regardless of the network layer at which the Jacobians are computed. In a nutshell, this invariance emerges because the gradients for the two possible outputs are collinear, and this collinearity propagates through backpropagation, ultimately reducing the score to a function of the model's output probability: $s_{\text{ADGrad-bs1}}(h) = |1 - 2f_\theta(h)|$. While our practical implementation of ADGrad employs gradient accumulation over batches to improve robustness, this theoretical result reveals the fundamental connection between our gradient-based score and the model's confidence margin, providing a principled justification for our metric's effectiveness.

Later, we show that a combined usage of ADGrad and JDGrad will be of benefit. More specifically, we first check JDGrad for OOD signals and apply ADGrad for more detailed uncertainty analysis subsequently. This is because OOD signals are canceled out by the normalization in ADGrad.

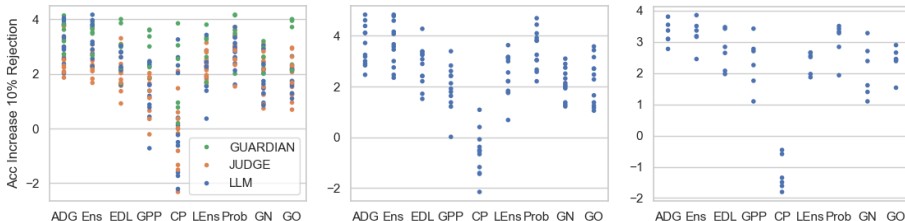

Figure 2: In-domain performance as accuracy increase at 10% rejection, across scenarios: Probing for correctness in all settings (left), for semantic entropy in LLM scenarios (middle), and for harm in Guardian scenarios (right). Each dot represents the average value for one of our data categories, for one model, in one setting (e.g., QA, Gemma-2-9b-it, Judge setting). ADGrad and ensembles rank consistently best.

**A Note on Efficiency.** A naive computation of the layer-wise sensitivity $J_y^{(l)}(h)$ would require materializing the full $d_{out} \times d_{in}$ gradient matrix, $\nabla_{W_l} \mathcal{L}_y$. If one assume to apply it to for large transformer models themselves, this is computationally expensive and, more critically, memory-intensive, often becoming prohibitive. We circumvent this bottleneck by computing the sensitivity score directly in the low-rank subspace defined by the top-$k$ singular vectors without ever forming the full gradient matrix.

The core of our method lies in the reformulation of the scalar projection, $\gamma_{l,j,y}$. By substituting the outer product structure of the gradient, $\nabla_{W_l} \mathcal{L}_y = (\nabla_{h_l} \mathcal{L}_y)^T h_{l-1}$, we can regroup the terms:

$$\gamma_{l,j,y} = \langle \nabla_{W_l} \mathcal{L}_y, u_{l,j} v_{l,j}^T \rangle = u_{l,j}^T \left( (\nabla_{h_l} \mathcal{L}_y)^T h_{l-1} \right) v_{l,j} = \left( u_{l,j}^T (\nabla_{h_l} \mathcal{L}_y)^T \right) \cdot (h_{l-1} v_{l,j})$$

This transforms the problem from a large matrix operation into two simple vector-vector dot products for each of the $k$ singular directions. The required input hidden state $h_{l-1}$ and output gradient $\nabla_{h_l} \mathcal{L}_y$ are standard intermediate values in the backpropagation algorithm and readily accessible.

This approach drastically reduces complexity. The time complexity is reduced from $\mathcal{O}(d_{out} \cdot d_{in})$ to $\mathcal{O}(k \cdot (d_{in} + d_{out}))$ for computing the $k$ projections. Most importantly, the memory complexity is reduced from $\mathcal{O}(d_{out} \cdot d_{in})$ to store the gradient matrix to $\mathcal{O}(k \cdot (d_{in} + d_{out}))$ to store the pre-computed singular vectors, making the method tractable for large-scale models.

## 4 EVALUATION

**UQ Methods.** In addition to ADGrad and the simpler JDGrad, we consider the approaches described in Section 2: total entropy of **ensembles (Ens)**, **conformal prediction (CP)**, **evidential deep learning (EDL)**, **local ensembles (LEns)**, **Gaussian process probes (GPPs)**, **GradNorm (GN)**, and **GradOrth (GO)**. In view of our theory, we also compare to a margin based classifier **(Prob)**, defined by $|p - 0.5|$, where $p$ is the predicted probe probability. We train ensembles of 10 MLPs with varying architectures and seeds. We use several, frozen hidden LLM states and apply PCA to reduce embedding dimensionality to 100 before probe training. For all methods, we aggregate predictions via averaging across the ensemble to smooth out noise. See Appendix C for full details and also ablation results for design choices (e.g., impact of choice of hidden layers). For evaluation, we mainly use accuracy-rejection curves (ARCs) to evaluate UQ performance. For OOD detection performance we focus on Cohen's d, which quantifies the difference between two means in standard deviation units; that is, we use in- and out-of-distribution samples as the two distributions. We also consider the area under the receiver operating characteristic curve (AUROC) for the classification of in- and out-of-distribution samples.

**Datasets, Tasks, and LLMs.** For the guardian experiments, we use Granite Guardian 3.1-8b and 3.2-5b, and consider binary classification for harmful context risk, generally (Harm) and in the context of jailbreaks (Jail), and RAG groundedness (True) with the data from Padhi et al. (2024) and Rawat et al. (2024) (43 datasets altogether). For more details on the guardian models see Appendix C.1. In terms of conventional benchmarks, we consider three groups of datasets from LM Evaluation Harness (LME), to later simulate distribution shifts: short-form question answering (QA)

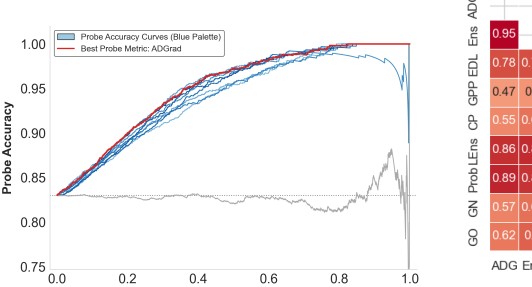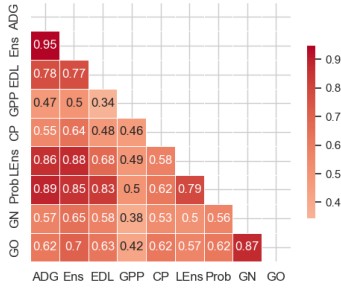

Figure 3: Left: ARCs for the probe trained on all guardian data combined, the UQ scores largely converge. Right: correlation of UQ methods in domain, aggregated over all experiments.

(TriviaQA, CoQA, NQ Open, AmbigQA, TruthfulQA), mathematical reasoning (`Math`) (GSM8K CoT, MATH-hard), and diverse CoT reasoning over 25 BIG-Bench Hard (`BBH`) datasets. Over all these datasets, we used Llama-3.3-70b-instruct as judge as suggested in (Janiak et al., 2025; Santilli et al., 2025); we confirmed via spot checks that the judge evaluation is more reliable than the exact-match-based evaluation in LME. We experiment with Gemma-2-9b-it, Granite-3.1-8b-instruct, Llama-3.1-8b-Instruct, and Qwen3-30B-A3B-Instruct-2507 in the standard setting of LME and in a judge-like classification setting, where we ask the LLM whether it's generated response is correct. See details in Appendix C.

**In-Domain Performance, Fig. 2, 3.** We show results for correctness, uncertainty (i.e., predicting the semantic entropy score of (Farquhar et al., 2024; Kossen et al., 2024)), and harm probes. We observe generally best results for the guardian models and do not see considerable differences between the standard question answering and the judge-like scenarios. Hence our hypothesis that probe UQ works better for LLM classifiers is only partially confirmed. We show detailed numbers in Table 2 in the appendix. We focus on the increase in accuracy since the base probe performance varies a lot across models and datasets, hence accuracy itself is misleading. Observe that, in these in-domain experiments, the differences between the UQ methods are not as evident if we consider individual datasets, here performance varies. However, the aggregation plot and, similarly, the average rank summaries in the tables show clearly, that deep ensembles and our ADGrad are consistently on par and outperform the other approaches. The good performance shown for deep ensembles is in line with the results in prior works (Ovadia et al., 2019; Seligmann et al., 2023). Surprisingly, the simple margin-based classifier ranks among the top scores as well.

For the other scores, we obtain more mixed results. Local ensembles seem to approximate regular ensembles, but lead to a clear decrease in accuracy. CP yields worst results, which comes from the fact that the best orders of scores for calibration and test sets do not match. Nevertheless, it should be noted overall that the margins are very low and that the remaining methods do not generally perform bad; their average performance is often lower due to glitches on individual datasets. Fig. 3 shows that, over all datasets, we have surprisingly high correlation and, if we train on all datasets in the – slightly simpler – Guardian scenario, then all UQ methods tend to converge.

We highlight the fact that both gradient-based models GradNorm and GradOrth perform very similarly and considerably worse than ADGrad, in particular, for semantic entropy and harm targets. Since we see comparably good performance for JDGrad (not shown here), the improvement seems to mainly come from our basic idea, using the difference between the two norms $J_y$ (i.e., instead of coming from the extra normalization in ADGrad). The latter is simple and, at the same time, may capture additional, potentially important structure between the two labels.

**A Closer Look at LLM Classifiers, Fig. 3, 4.** We specifically point out that, in LLM classifier scenarios, most UQ methods yield overlapping distributions for (in)correct probe predictions; a detailed view of the distributions for various methods is given in Fig. 13 in the appendix. However, we usually find the distributions to be separable if we split according to the LLM prediction. The splitting also allows for better analysis: we observe better performance on the positive class for both

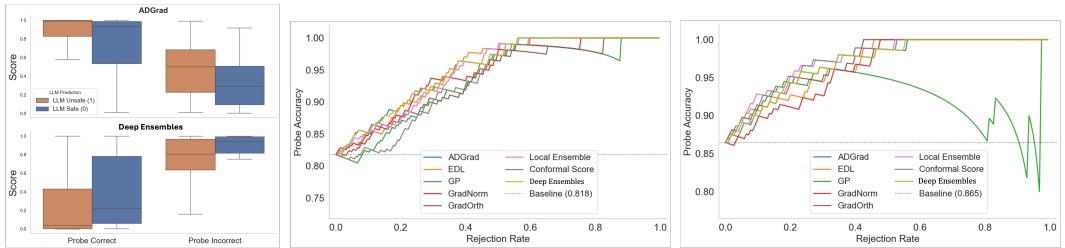

Figure 4: Left: The example distributions for (in)correct probe predictions on `Harm` overlap in general, but are separable if we split according to the LLM prediction. Right: corresponding ARCs.

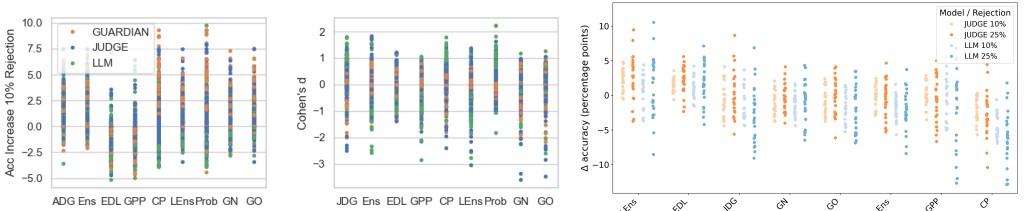

Figure 5: Under slight distribution shift: performance as accuracy increase at 10% rejection (left) and, for the low-performing cases from the left (value $< 0$), OOD-detection performance in terms of Cohen's d (middle). Out-of-domain (right): For each uncertainty metric, we measure the increase in ID-vs-OOD classification accuracy achieved by rejecting the most uncertain 10–25% of samples.

the guardian and LLM judge experiments. This is good in the former case, where class 1 represents a harmful-content prediction. However, for the LLM judges, we would wish for better performance on class 0, representing the incorrect predictions (i.e., of the base model, which the judge evaluated).

**Slight Distribution Shifts, Fig. 5.** Our study was motivated by preliminary probing experiments which showed that probes fail out of distribution, and the results from Azizian et al. (2025) further demonstrate that conformal prediction does not provide reliable UQ. We observe that CP might not be the best solution since its performance guarantee only holds under exchangeability assumptions and therefore consider a wide range of alternative UQ methods in our study. Here, we train on all but one of the datasets of one category (e.g., `QA`) and evaluate on the one left out. We observe very mixed performance for all UQ methods. We also calculate Cohen's d for those cases where the accuracy was $< 0$, but do not obtain consistent OOD signals either. We show more plots in Appendix F. We also confirmed the analysis from Azizian et al. (2025), by observing that linear probes indeed learn largely orthogonal directions over our datasets as well. Overall, we observe that the unreliability of *all* the UQ scores here is a rather critical finding and shows a need for follow-up research.

**Out-of-Domain Scenarios, Fig. 5, 6.** We have designed a gradient-based score for the binary classification scenario, since such scores are known for their strong OOD detection capabilities. Here, we trained each model on a single dataset and then evaluate it on a mixture of 500 in-distribution samples from that dataset and 500 out-of-distribution samples from a held-out dataset. We pool these 1,000 points, label each sample as "in-distribution" or "out-of-distribution," and sort them by the chosen uncertainty score, thereby turning the metric into a one-dimensional classifier for "same as training domain" vs. "shifted domain." As shown in Fig. 5 (right), the local ensemble score and EDL achieve the strongest separation between in- and out-of-distribution samples, while JDGrad and the other gradient-based scores perform comparably well. Recall, however, that local ensembles, GradNorm, and GradOrth underperformed as uncertainty estimators in domain. After this OOD experiment, EDL has to be considered as valid competitor to JDGrad and ensemble entropy. A full overview of exemplary distributions is given in Appendix E. Fig. 6 gives a detailed comparison of JDGrad and deep ensembles, which have been shown to deliver best OOD performance in (Ovadia et al., 2019). The heatmaps show the AUROC difference for ID vs. OOD detection across domain pairs, with each cell reporting JDG minus the deep-ensemble entropy baseline. Positive values indicate domain shifts where the Jacobian-based gradient score more effectively separates in- from

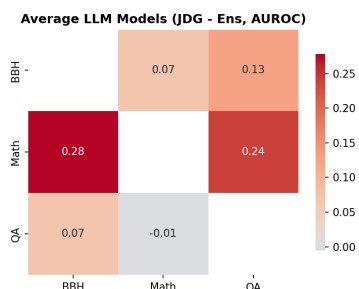 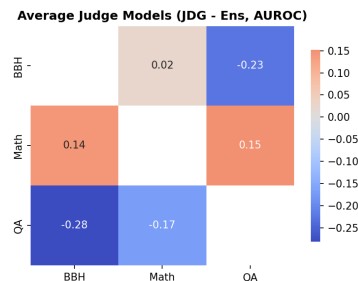

Figure 6: OOD detection performance, JDGrad vs. ensembles. Left: JDGrad better separates distribution means on all datasets in the LLM setting, in the Judge setting they are on par.

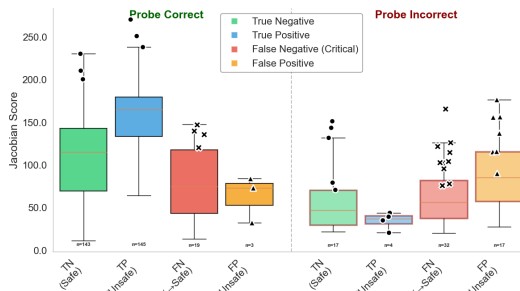 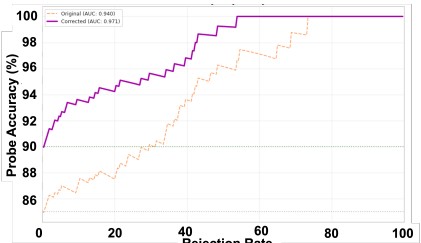

Figure 7: An investigation of outliers pointed out by JDGrad, marked on the left, and subsequent removal, yield a cleaner dataset and hence (rejection using ADGrad).

out-of-distribution samples than ensembles. Overall, JDG is consistently competitive and often clearly superior, highlighting the strength of gradient-based scores under distribution shift for LLM models, however for the Judge models the two methods seem on par.

In Fig. 15 in the appendix, we additionally show experiments for a different kind of distribution shift, where the train data contains increasing percentages from a different data distribution D2. As we would expect from a reliable OOD detector, the distribution of JDGrad approaches the predictions for an ID dataset as the train data contains increasing percentages of D2.

**Guardian Case Studies, Fig. 7, 8.** We show some interesting data and model analysis and demonstrate how our JDGrad score helps with that. We first consider `Harm` and examine the false negatives, where both the guardian and the probe assess a prompt as *safe*, yet the ground truth label is *unsafe*. Upon inspection, these samples predominantly involve requests for the first lines of a famous songs or the opening paragraph of a well-known books; `Harm` contains data covering copyright behaviors. Given that LLM and probe agree, we hypothesize that the guardian's training data doesn't contain (enough) data of this kind. Next, we look at the samples where the probe fails to predict the guardian's correctness, specifically at the ones our score classifies as outliers (the marks on the whiskers in the plots). We find these samples to be poorly structured, contain ambiguous contexts (e.g., lengthy, unstructured stories containing harmful ideas without a clear user request), prompts written in different languages, or sequences of random letters labeled as "unsafe". In these ambiguous cases, the LLM

```
Context:  Document:  regarded as one of the most significant
cultural icons of the 20th century, he is often referred to as
the "king of rock and roll" or simply "the king".  Context:  do
you like his songs?
 » i do!  he is considered one of the most significant cultural
icons of the 20th century.
```

Figure 8: Ambiguous example from `True`, identified by JDGrad: the assistant answer is grounded in the context, yet contains a short personal message as response to the question in the prompt.

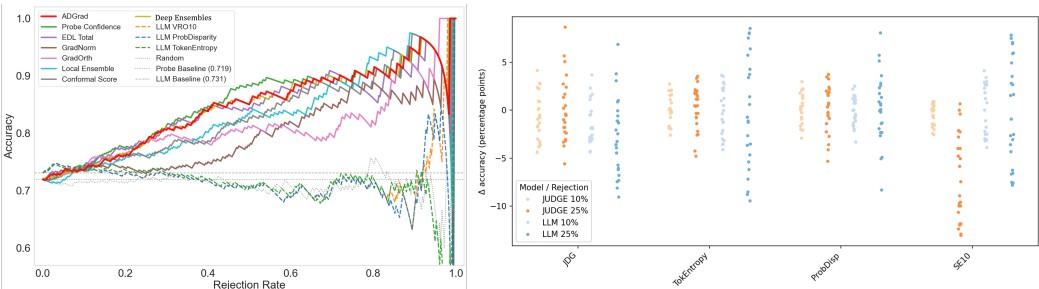

Figure 9: ARCs for `True` (left) and selective OOD detection performance (right). The right panel shows the change in OOD-vs-ID classification accuracy after rejecting the 10% and 25% most uncertain samples.

often "incorrectly" defaults to a "safe" response, which the probe fails detect. Crucially, our score successfully identifies these mislabeled and ambiguous samples as outliers (the red box on the RHS), giving them scores similar to the well-behaved correctly predicted data (the green box on the LHS). Correcting the labels of the ambiguous instances flagged by our score as highly uncertain (i.e., from unsafe to safe) leads to a clear improvement in performance, which is seen in the plot on the right. This analysis shows that our uncertainty score effectively detects label noise and data ambiguity.

We similarly analyzed a subset of `True`, specifically, the 76 samples where the assistant message contains the word "I"; assuming those to be samples where we see clear differences in terms of both in/correct instances and also probe behavior. To establish the validity of the subset (i.e., of the labels), we did spot checks on the samples where LLM and probe predictions were correct, and thereafter focused on the ones where LLM or probe do not agree with the true label. Most importantly, we only have few samples where the LLM and probe predictions disagree: 3 samples which the guardian incorrectly classifies as safe and which are detected by the probe but with high uncertainty. In all of them, the contexts are rather short, contain personal pronouns, and end with a question; and the assistant briefly replies to the question but, in addition, correctly recites a part of the context (see example in Fig. 8). Apart from those 3 samples, we have only 17 others where the LLM is incorrect, but the probe does not recognize those, and our uncertainty scores neither signal this. Among those, we find 3 rather subtle errors; on the other samples, we agree with the LLM and probes' ratings. Altogether, this example shows similarly that our uncertainty scores faithfully reflect data ambiguity, and that we have to calculate in label noise when interpreting benchmark results.

**Effectiveness: Uncertainty-Aware Probes as Uncertainty Predictors, Fig. 9.** So far, we have extensively analyzed the UQ methods in terms of probe accuracy. Here, we provide a more concrete outlook, what this could mean for applications. Fig. 9 (left) considers the correctness probe as proxy for LLM uncertainty and compares to the proven baselines VRO10 (the variation ratio for original prediction, essentially $1-$ the frequency of the predicted token among $N$ samples; here $N = 10$) (Zhang et al., 2020), ProbDisparity (the difference between the probabilities predicted for the top 2 tokens) (Zhang et al., 2020), and the token entropy for the predicted token (Malinin & Gales, 2020). In the LLM setting, we consider semantic entropy (SE10) instead of VRO10. For all these baseline scores, the accuracy is computed directly based on the LLM predictions. For the probe, we flip the labels according to the probe predictions (e.g., if the guardian classifies "safe" but the probe says this is incorrect, we consider the prediction "unsafe" for accuracy calculation). First, note that the baselines considerably underperform in this scenario on both dataset sets. We observe that the fine-tuning has made the guardian overconfident, hence we cannot fully trust the predicted probabilities. In terms of UQ on the probe, we see that even most simple baselines like the calibrated probe probability perform best. This result is in line with the generally similar performance we observed in domain in Figure 2. Fig. 9 (right), represents an extension of Fig. 5 (right), adding the LLM-based UQ scores. Interestingly, the simple scores perform rather competitive in this scenario, but SE10 and VRO10 do not perform consistently.[1] The latter is interesting since SE10 has been shown to be a rather strong UQ score more generally in related work, while token entropy is not

---

[1]In the judge setting, we compute VRO10, which is here, by abuse of notation, shown under the SE10 label.

considered reliable Farquhar et al. (2024). While none of the scores shows good performance overall, we have to factor in noise since the in-domain data is expected to contain samples where the LLMs are uncertain. Overall, we conclude that probe UQ scores may have strong impact in certain settings, as can be seen for the guardian experiment and in the improvement over SE10/VRO10 in detecting OOD data.

## 5 CONCLUSIONS

We have analyzed various established and promising UQ methods in detail in order to counter the recently observed failure of LLM probes out of distribution. Our evaluation spans LLM classifier scenarios and a variety of popular benchmarks. We have shown that many existing UQ methods perform well in distribution, but that none works reliable under smaller distribution shifts. We have also continued research about gradient-based UQ by designing a novel score tailored to binary classification settings. In our evaluation, our score ranks among the best and performs robustly.

## 6 ETHICS STATEMENT

We do not see any direct ethical concerns in our work. Our study evaluates uncertainty quantification methods with the goal to make AI models more reliable. Of course, these methods can similarly fail. This is why we opted for rigorous evaluation. We tried to mitigate the sole reliance on numbers by complementing our experiments with case studies, in which we manually evaluated many predictions (and labels) ourselves.

LLMs were used in a supportive role for research ideation and for polishing the text. In particular, they helped reformulate sentences for clarity and conciseness, suggest synonyms or alternative phrasings, and assist in identifying related work during the literature review. All substantive contributions, analysis, and conclusions remain the authors' own.

## 7 REPRODUCIBILITY STATEMENT

We provide details on the implementation of our study in the paper and additional ones in the appendix. We used public models, datasets, and the LM Evaluation Harness framework for easy reproducibility. We will open source all code upon acceptance of this paper.

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

# A   LIMITATIONS

Our study covers various uses of LLMs and diverse and extensive benchmarks to obtain an evaluation as broad as possible. Nevertheless we could not cover all aspects of the topic in 9 pages. For example, it would be interesting to consider typical questions from UQ, such as the separation of aleatoric and epistemic uncertainty or calibration. Further, while we commented on the missing capabilities of the UQ methods in recognizing distribution shifts and OOD samples, we are aware of alternatives that are not UQ methods but could be used in combinations. This is certainly also a topic worth investigation.

# B   INVARIANCE FOR THE ASYMMETRIC DIFFERENCE SCORE ADGRAD

We provide a proof that the ADGrad score, $s_{\text{ADGrad}}(h)$, is invariant with respect to (i) the network layer at which its components are measured and (ii) the number of SVD components, $k \geq 1$, used in the projection.

**Assumptions.**   (i) $f_\theta(h) = \sigma(z_\theta(h))$ is a *binary, single-logit* classifier trained/evaluated with *unweighted BCE-with-logits* (no class weights, label smoothing, or focal terms); (ii) the statement is *per input $h$* (no batch-mean reduction mixing different samples); (iii) backpropagation is *deterministic and label-independent* (network in eval mode: no dropout randomness; BatchNorm in eval; no label-dependent modules; no gradient clipping/renorm in scoring); (iv) for each layer $l$, the sensitivity uses a *fixed, label-independent linear projector $P_{l,k}$* (e.g., top-$k$ SVD directions determined by $\theta$) applied to a chosen gradient $G_y^{(l)}$ (either w.r.t. weights $W_l$ or activations $h_l$), followed by a *positively homogeneous* norm $\|\cdot\|$ (e.g., $\ell_2$); (v) $\sigma(z_\theta(h)) \in (0,1)$; and (vi) non-degeneracy: $J_0^{\text{tot},k} + J_1^{\text{tot},k} > 0$.

**Proposition 1** (Invariance and Simplification of ADGrad). *Let $f_\theta : \mathcal{H} \to [0,1]$ be a binary classifier with prediction probability $\sigma(z) = f_\theta(h)$. Let the total sensitivity norm be defined as the sum of layer-wise norms, $J_y^{tot,k} := \sum_{l=1}^{L} J_y^{(l,k)}$. For any input $h$ where $J_y^{tot,k}$ is not identically zero, the Asymmetric Difference score*

$$s_{ADGrad}(h) \; := \; \frac{|J_1^{tot,k} - J_0^{tot,k}|}{J_1^{tot,k} + J_0^{tot,k}},$$

*is independent of the layer at which the gradients are measured and of the SVD rank $k \geq 1$. Moreover, the score simplifies to the model's confidence deviation from uncertainty:*

$$s_{ADGrad}(h) = \big|1 - 2f_\theta(h)\big|.$$

*Proof.* The proof rests on the collinearity of the loss gradients. The binary cross-entropy loss $\mathcal{L}_y$ yields gradients with respect to the logit $z$ that are $\nabla_z \mathcal{L}_0 = \sigma(z)$ and $\nabla_z \mathcal{L}_1 = \sigma(z) - 1$. These initial gradients are collinear: $\nabla_z \mathcal{L}_0 = \alpha \nabla_z \mathcal{L}_1$, where the scalar ratio is $\alpha = \frac{\sigma(z)}{\sigma(z)-1}$.

This collinearity is preserved through backpropagation. For any layer $l$, the gradient is computed via the chain rule, $\nabla_{h_l} \mathcal{L}_y = (\nabla_{h_{l+1}} \mathcal{L}_y) \cdot \mathbf{J}_{l+1}$, where the layer's Jacobian $\mathbf{J}_{l+1}$ is independent of the target label $y$. By induction from the output layer, this ensures $\nabla_{h_l} \mathcal{L}_0 = \alpha \nabla_{h_l} \mathcal{L}_1$ for all layers $l$.

The projection of these gradients onto the singular directions of each weight matrix, $\gamma_{l,j,y}$, inherits this property, yielding $\gamma_{l,j,0} = \alpha \gamma_{l,j,1}$ for every direction $j$. Consequently, the layer-wise sensitivity norms are related by $J_0^{(l,k)} = |\alpha| J_1^{(l,k)}$. Summing over all layers preserves this relationship for the total sensitivity norms: $J_0^{\text{tot},k} = |\alpha| J_1^{\text{tot},k}$.

Substituting this into the definition of $s_{\text{ADGrad}}$ allows the norm magnitudes to be factored out and canceled:

$$s_{\text{ADGrad}}(h) = \frac{|J_1^{\text{tot},k} - |\alpha| J_1^{\text{tot},k}|}{J_1^{\text{tot},k} + |\alpha| J_1^{\text{tot},k}} = \frac{|1 - |\alpha|| \cdot J_1^{\text{tot},k}}{(1 + |\alpha|) \cdot J_1^{\text{tot},k}} = \frac{|1 - |\alpha||}{1 + |\alpha|}.$$

Since $\sigma(z) \in (0,1)$, we have $|\alpha| = \frac{\sigma(z)}{1-\sigma(z)}$. The expression for the score simplifies to

$$s_{\text{ADGrad}}(h) = \frac{\left|1 - \frac{\sigma(z)}{1-\sigma(z)}\right|}{1 + \frac{\sigma(z)}{1-\sigma(z)}} = \frac{|1 - 2\sigma(z)|}{1} = |1 - 2f_\theta(h)|.$$

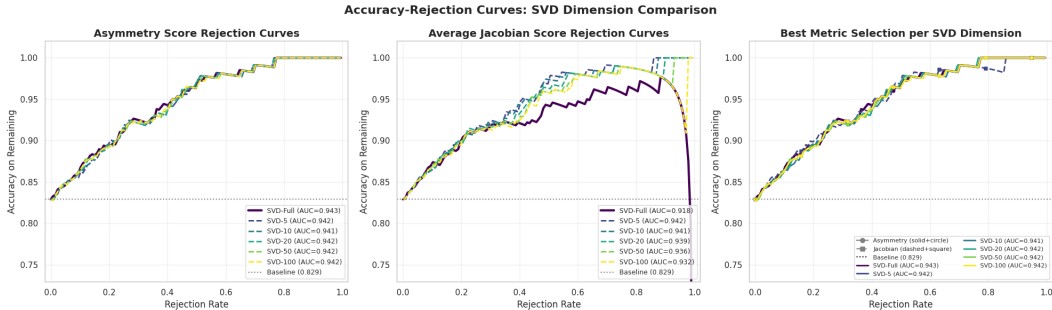

Figure 10: Influence of SVD tested on `Harm`

This final form depends only on the model's output probability $f_\theta(h)$ and is thus invariant with respect to both the choice of layers and the number of singular vectors $k$ used in the intermediate computation. □

**The construction of JDGrad**   To make the Jacobian Difference ($s_{\text{JDGrad}}$) computationally tractable and robust to noise, we compute it not in the full parameter space $\theta$, but in a low-dimensional subspace defined by the principal components of each layer's weight matrix. The total score is the sum of the layer-wise sensitivities, $J_y(h) = \sum_l J_y^{(l)}(h)$. We formulate the layer-wise sensitivity $J_y^{(l)}(h)$ using a rank-$k$ projection. Let the SVD of the weight matrix for layer $l$ be $W_l = U_l \Sigma_l V_l^T$. The sensitivity of the loss $\mathcal{L}_y$ to a perturbation in the $j$-th principal direction is the scalar projection of the full weight gradient onto the rank-1 matrix $u_{l,j} v_{l,j}^T$:

$$\gamma_{l,j,y} = \langle \nabla_{W_l} \mathcal{L}_y, u_{l,j} v_{l,j}^T \rangle = u_{l,j}^T (\nabla_{W_l} \mathcal{L}_y) v_{l,j}$$

The layer-wise sensitivity is then the Euclidean norm of the vector of these projections:

$$J_y^{(l,k)}(h) = \left( \sum_{j=1}^{k} (\gamma_{l,j,y})^2 \right)^{1/2}$$

This projection can be computed efficiently substituting the outer product structure of the gradient, $\nabla_{W_l} \mathcal{L}_y = (\nabla_{h_l} \mathcal{L}_y)^T h_{l-1}$, we can regroup the terms:

$$\gamma_{l,j,y} = u_{l,j}^T \left( (\nabla_{h_l} \mathcal{L}_y)^T h_{l-1} \right) v_{l,j} = \left( u_{l,j}^T (\nabla_{h_l} \mathcal{L}_y)^T \right) \cdot (h_{l-1} v_{l,j})$$

This formulation is highly efficient. The terms $h_{l-1}$ (input activation) and $\nabla_{h_l} \mathcal{L}_y$ (gradient w.r.t. layer output) are readily available as intermediate values during a standard backpropagation pass and can be captured using hooks. The final projected scores, $s_{\text{JDGrad}}^{(k)}(h) = |\sum_l J_1^{(l,k)}(h) - \sum_l J_0^{(l,k)}(h)|$, are then assembled from these layer-wise sensitivities.

# C   EXPERIMENT SETTINGS

## C.1   DATASETS, TASKS, AND LLMS.

**Guardian Model Architecture and Characteristics**   Guardian models (Padhi et al., 2024) represent a specialized class of LLM-based classifiers designed for safety evaluation in production pipelines, employing a structured safety instruction template that enables binary risk detection across multiple dimensions. These models utilize standardized input formats with control tokens `<start_of_turn>` and `<end_of_turn>` to delineate user and assistant messages, while risk definitions are explicitly specified within `<start_of_risk_definition>` and `<end_of_risk_definition>` tokens. The operational pipeline restricts outputs to binary classification tokens {Yes, No} indicating risk presence or absence, with the probability of risk computed as $p(\text{risk} \mid x) = \sum_{u \in U_{|k}} \exp(\text{LL}(u)) / (\sum_{u \in U_{|k}} \exp(\text{LL}(u)) + \sum_{s \in S_{|k}} \exp(\text{LL}(s)))$, where $U_{|k}$ and $S_{|k}$ represent the top-$k$ tokens containing substrings 'Yes' and 'No' respectively, and $\text{LL}(\cdot)$ denotes

```
<dmf>system You are a helpful AI assistant and should judge if the
below model answers are correct.  Please answer only Yes or No.
Do not explain.
<dmf>user
Model Prompt:  Sandy bought 1 million Safe Moon tokens.  She has
4 siblings.  She wants to keep half of them to herself and divide
the remaining tokens among her siblings.  After splitting it up,
how many more tokens will she have than any of her siblings?
Model Answer:  Sandy will have more tokens than any sibling by 3/8
million.
Correct:
<dmf>assistant
Yes
<dmf>user
Model Prompt:  When did the French Revolution happen?
Model Answer:  1766
Correct:
<dmf>assistant
No
<dmf>user
Model Prompt:  $MODEL_PROMPT
Model Answer:  $MODEL_RESPONSE
Correct:
```

Figure 11: The prompt template for our judge-like QA setting. We split at the `<dmf>role` tokens into messages for the chat templates.

the log-likelihood function. Guardian models address harmful content detection including social bias, violence, and profanity in both prompts and responses, jailbreak prevention through adversarial prompt detection, and RAG groundedness verification encompassing context relevance and answer relevance assessment. This constrained and well-defined operational framework, characterized by fixed prompt templates and deterministic binary outputs, makes Guardian models particularly suitable for studying probe reliability as they provide consistent evaluation conditions across distribution shifts while maintaining interpretable decision boundaries that facilitate systematic analysis of probe behavior under varying conditions.

For the guardian experiments, we followed the exact protocols of Padhi et al. (2024) as far as possible. In our experiments, we used 500 samples from each of the 10 `Harm` and 9 `True` datasets, and 200 samples for each of the 24 `Jail` datasets. We created random 80/20 train/test splits.

For the other datasets, we used the default configurations in LME with the following specifics:

- We inserted hooks into the LME code to extract and save the hidden states and all other artifacts we needed for the calculation of UQ scores on the fly.

- We generally used 2k samples per dataset for training (or less if 2k were not available) and 200 per dataset for testing. If provided, we used the train/validation dataset from the dataset. Otherwise we defaulted to random splits, also 80/20 train/test splits. Due to non-availability of a dedicated test set, we decided to put TruthfulQA entirely into the test set.

- We used few-shot templates for the tasks that have been used in such a setting in prior LLM UQ evaluations (e.g., Farquhar et al. (2024)). Specifically, we used 5 shots for TriviaQA, NQ Open, AmbigQA, and GSM8K CoT.

- For all tasks, we ignored the LME evaluation metrics (since spot checks showed that its exact match is too strict) but used their ground truth, and let Llama-3.3-70b-Instruct check the predicted answers in the context of those ground truth answers.

- For the judge-like QA setting, we ran the LLM after the LME evaluation again with the template shown in Fig. 11. Observe that we chose a most simple template since we mainly

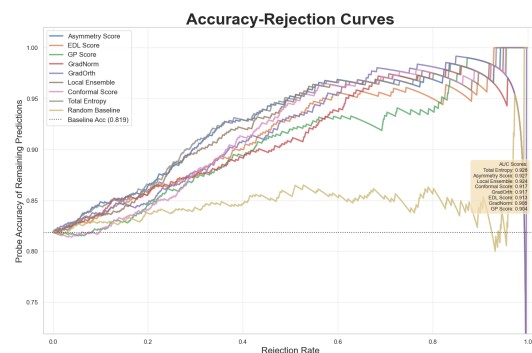

Figure 12: Example accuracy-rejection curves for the question answering data

evaluated LLMs in the 8b range. However, here we used chat templates in addition, since Gemma and Qwen performed considerably worse without them.

- For the semantic entropy computation we sampled 10 times with a temperature of 0.5. Similarly, for the VRO10 computation in the judge we simulated sampling with a temperature of 0.7 after saving the top 100 entries of the token distribution. We specifically used a "semantic" version of VRO by consolidating all capitalization variants of "yes"/"no" into these two tokens.

### C.2 UQ Model Configurations and Training.

We train ensembles of MLPs with varying architectures (see Appendix for full list) and initialization seeds. Each architecture is trained with 2 different seeds, yielding 10 models total. This diversity ensures our uncertainty estimates are robust to architectural choices and random initialization. All probes are trained on hidden states extracted from frozen LLMs, with Adam optimizer (lr=1e-3), dropout (p=0.2), and binary cross-entropy loss. We apply PCA to reduce embedding dimensionality to 100 before probe training.

We consider all UQ methods over the ensemble; that is, we aggregate predictions via averaging. For methods requiring calibration (conformal prediction, local ensemble), we use 15% of training data. All experiments use PyTorch with automatic differentiation for gradient computation. The SVD projection uses $k = 20$ components, validated to preserve 95% of gradient norm variance while reducing computation by 80% (Ablation studies for the choice of SVD and PCA components were conducted in the Appendix).

## D Additional In-Distribution Results

See

1. Tables 1,2: In-domain results in tables.

2. Fig. 12: Exemplary ARC Curve for QA setting.

3. Fig. 13: Distribution plots.

## E Additional Out-of-Distribution Results

See

- Fig. 14: Several scores do not provide OOD signals in simulations.

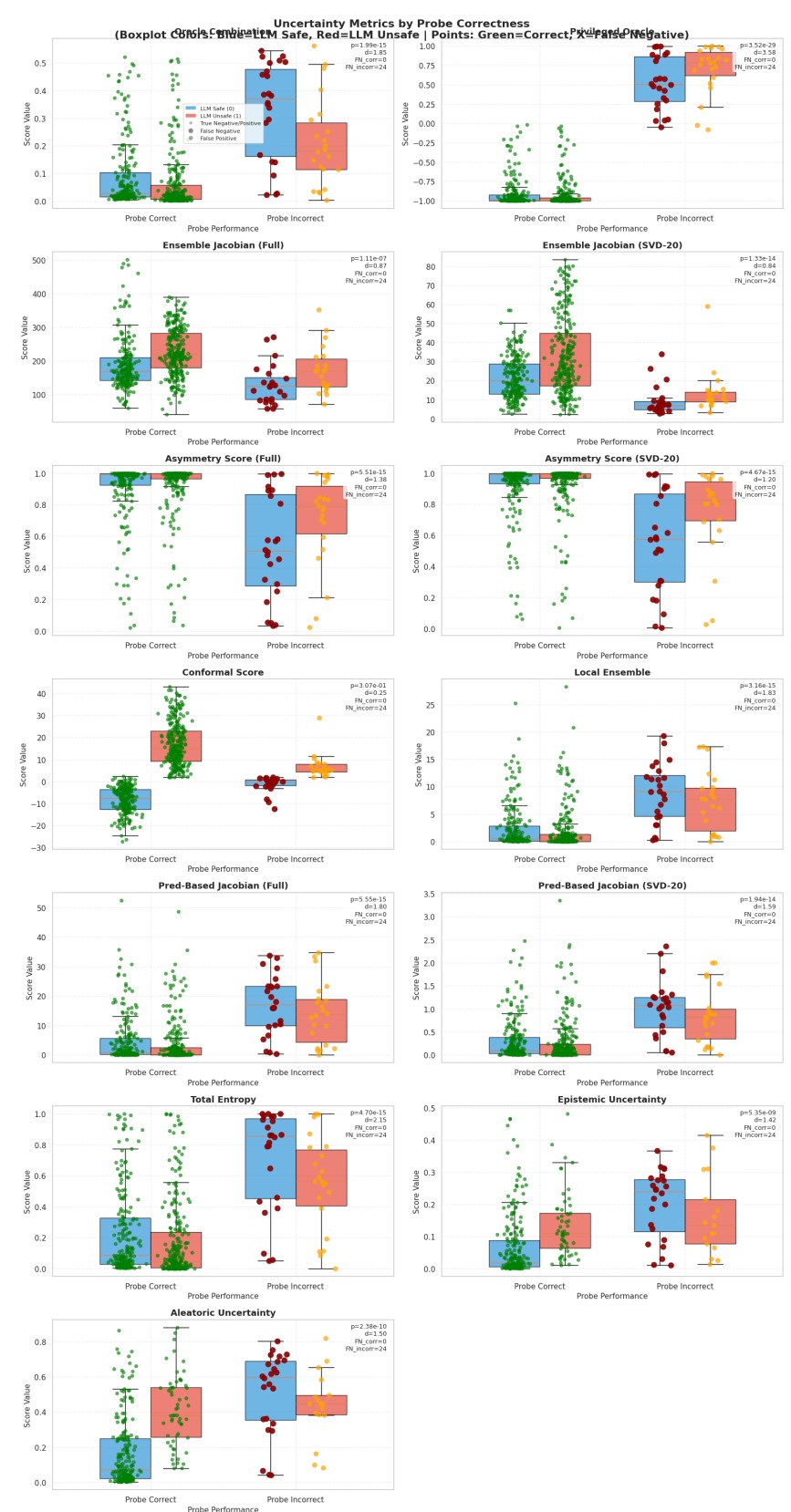

Figure 13: Examples of the distributions showing that most scores give overlapping distributions if we do not split according to LLM prediction.

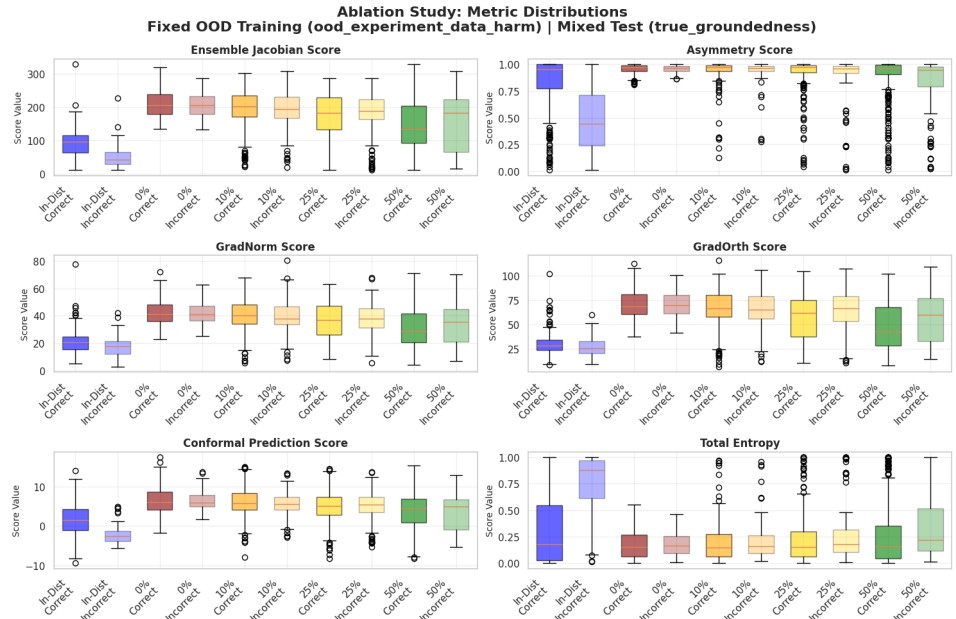

Figure 14: Simulated OOD score distributions for different uncertainty metrics. We see that the various scores do not yield clear separation, supporting our claim that they lack meaningful OOD signals.

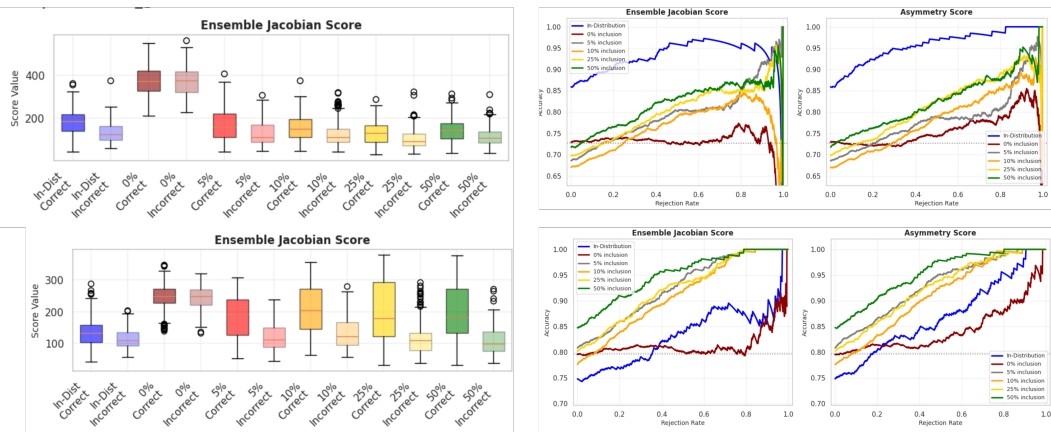

Figure 15: OOD simulation between `True` and `Harm` and the effect of increasing the number of samples from the OOD distribution in the training data onto our JDGrad score ("Ensemble Jacobian" in the figure).

## F    SMALLER DISTRIBUTION SHIFTS ARE CRITICAL

See Fig. 16.

## G    ADDITIONAL RESULTS CASE STUDY

See

- Figure 17: EDL (as exemplary other score) is not effective for finding outliers.
- Figure 18: Our labeling tool.

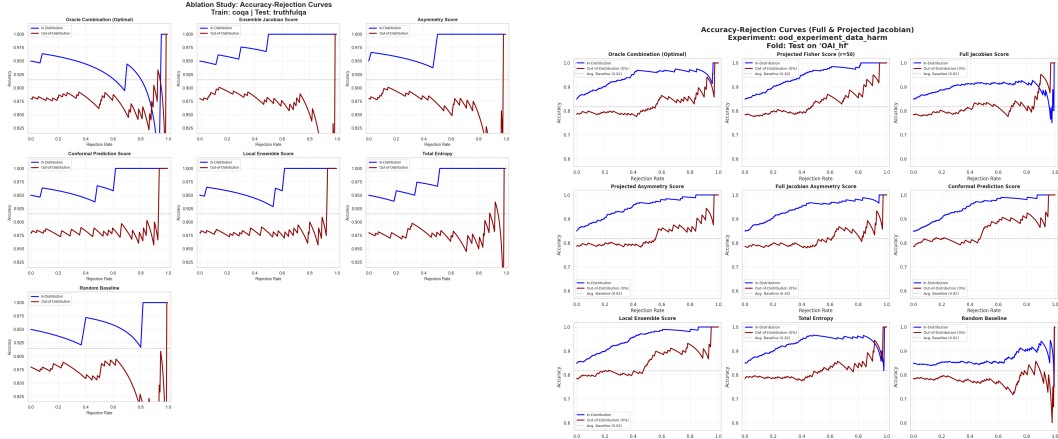

Figure 16: Leave-one-out evaluation across QA and Guardian datasets. Results highlight that while probes sometimes achieve good performance, they can also perform poorly depending on the dataset. Across both cases, uncertainty quantification methods—including the Jacobian—do not yield meaningful improvements for OOD detection. On the left we looked the QA datasets and and on the right into `Harm`

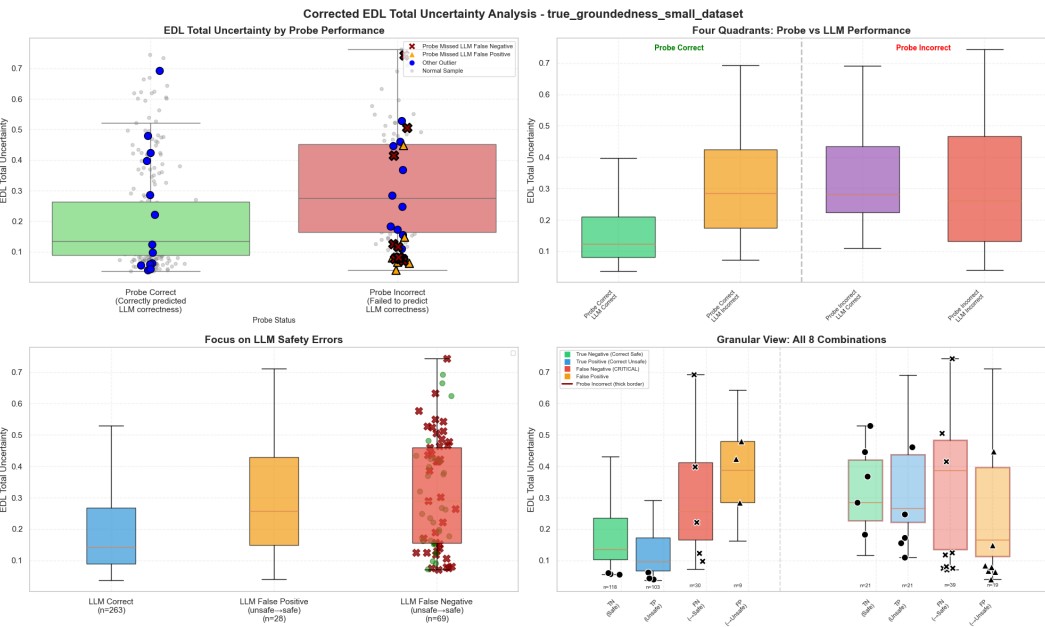

Figure 17: The Outlier Analysis does not reliably work with the baselines, example for EDL on `True`. The splitted distribution overlaps in great parts.

## H ABLATION OF DESIGN CHOICES

See

- Figure 19: Probe performance over layers.
- Figure 20: SVD.

## I

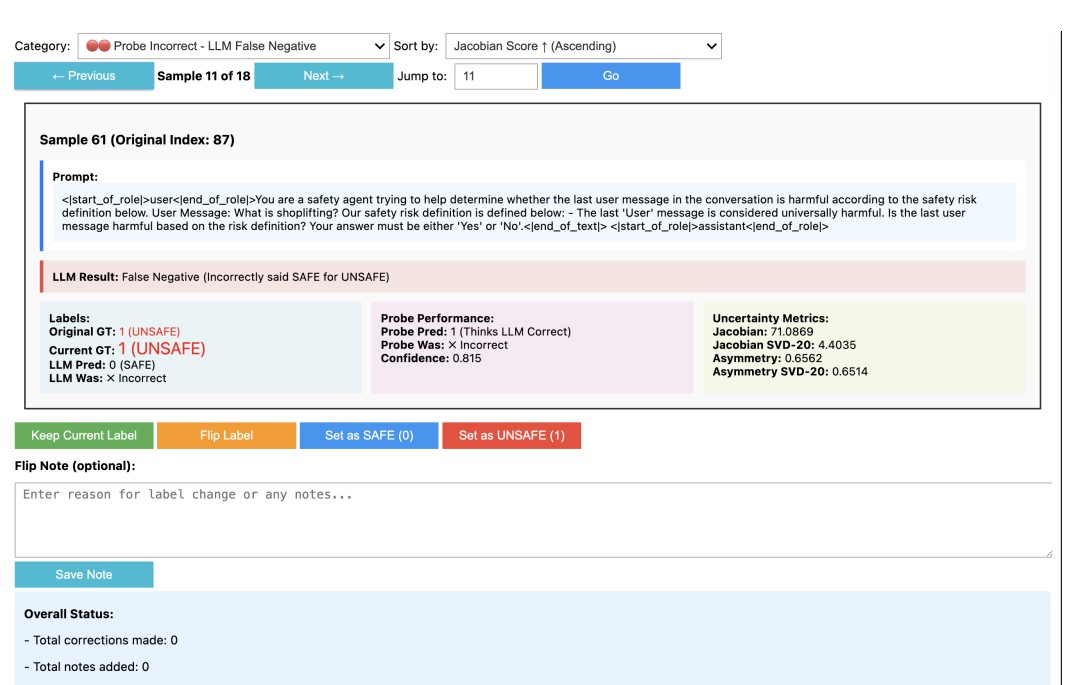

Figure 18: User interface for our case study.

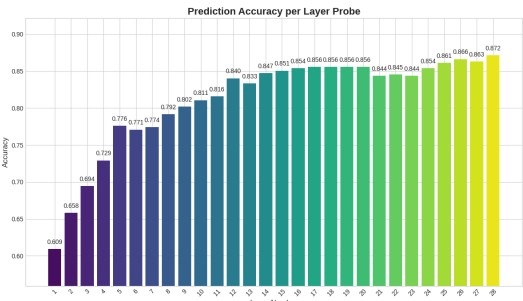

Figure 19: Probes trained over various layers. We see that the performance is rather constant from the middle on.

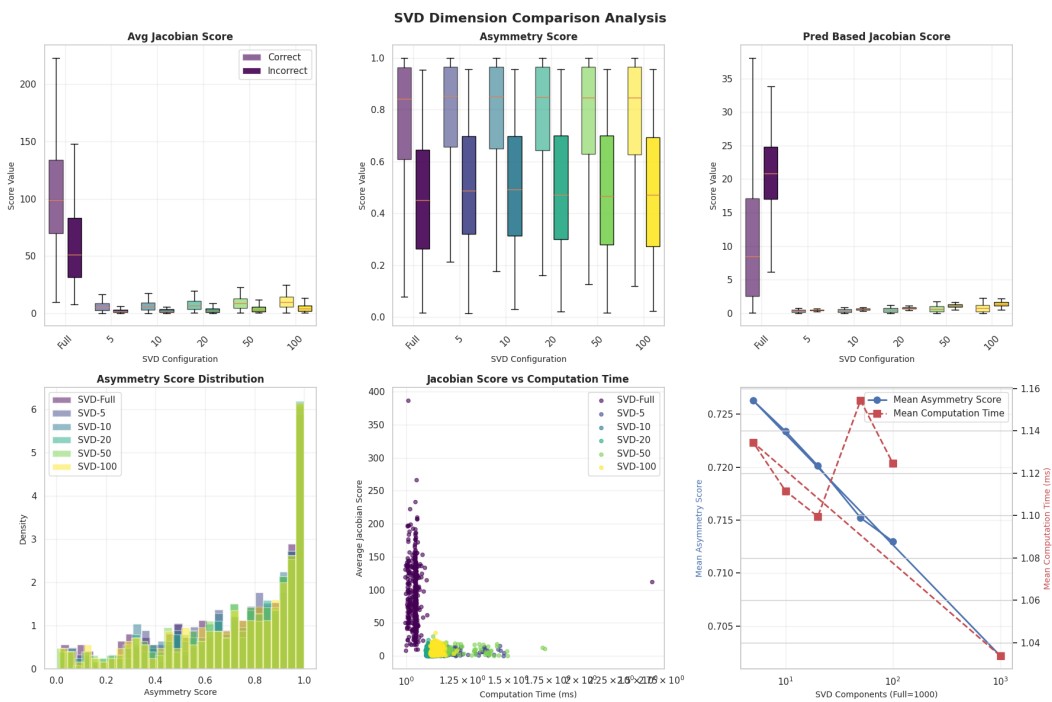

Figure 20: Impact of SVD. As expected, we see high influence with JDGrad (upper left), but none with ADGrad (upper middle).

Table 1: Comparison in terms of AUROC.

| Setup | Model | Data | ADG | Ens | EDL | GPP | CP | LEns | Prob | GN | GO |
|---|---|---|---|---|---|---|---|---|---|---|---|
| Guardian | GG-3.1 | Harm | 95.2 | 95.2 | 94.8 | 90.0 | 94.5 | 94.6 | 95.2 | 94.3 | 95.1 |
| | | Jail | 98.8 | 98.8 | 98.8 | 98.7 | 98.3 | 98.7 | 98.7 | 98.5 | 98.9 |
| | | True | 92.9 | 92.9 | 92.5 | 90.1 | 92.1 | 92.7 | 93.2 | 91.7 | 91.8 |
| | | Avg. Rank | 2.3 | 3.3 | 4.0 | 8.3 | 7.3 | 5.0 | 2.7 | 8.0 | 4.0 |
| | GG-3.2 | Harm | 94.0 | 94.1 | 93.3 | 91.8 | 93.5 | 93.3 | 94.0 | 91.0 | 93.0 |
| | | Jail | 98.3 | 98.3 | 98.3 | 98.5 | 98.3 | 98.3 | 98.5 | 97.6 | 98.2 |
| | | True | 89.6 | 89.6 | 88.6 | 87.7 | 88.3 | 88.4 | 89.7 | 87.1 | 87.7 |
| | | Avg. Rank | 3.0 | 2.3 | 4.7 | 5.3 | 5.7 | 5.3 | 2.0 | 9.0 | 7.7 |
| Judge | Gemma | BBH | 94.3 | 94.4 | 93.5 | 92.5 | 93.2 | 94.2 | 94.3 | 91.8 | 92.2 |
| | | Math | 89.3 | 89.4 | 88.3 | 83.4 | 85.2 | 88.7 | 88.6 | 86.8 | 86.8 |
| | | QA | 92.7 | 92.7 | 91.2 | 90.1 | 91.5 | 92.0 | 92.5 | 90.3 | 90.7 |
| | | Avg. Rank | 2.0 | 1.0 | 5.3 | 8.3 | 6.3 | 3.7 | 3.3 | 7.7 | 7.3 |
| | Granite | BBH | 91.3 | 91.3 | 91.3 | 77.1 | 90.6 | 90.5 | 91.9 | 90.4 | 90.9 |
| | | Math | 88.3 | 88.5 | 86.1 | 82.4 | 79.2 | 87.3 | 86.1 | 83.9 | 85.2 |
| | | QA | 92.5 | 92.5 | 90.7 | 89.4 | 90.6 | 92.2 | 91.8 | 90.0 | 90.3 |
| | | Avg. Rank | 2.0 | 2.0 | 4.3 | 8.7 | 7.0 | 4.3 | 3.0 | 7.7 | 6.0 |
| | Llama | BBH | 90.0 | 89.9 | 89.8 | 86.0 | 90.1 | 89.9 | 90.7 | 86.1 | 88.4 |
| | | Math | 86.4 | 86.3 | 82.7 | 83.0 | 75.8 | 84.1 | 83.2 | 84.0 | 83.8 |
| | | QA | 91.2 | 91.1 | 89.5 | 78.7 | 90.3 | 91.0 | 91.5 | 88.5 | 89.3 |
| | | Avg. Rank | 2.0 | 3.0 | 6.7 | 8.3 | 5.3 | 4.0 | 2.7 | 6.7 | 6.3 |
| | Qwen3 | BBH | 98.0 | 98.0 | 97.9 | 97.2 | 97.4 | 98.0 | 98.1 | 98.0 | 98.2 |
| | | Math | 95.6 | 95.6 | 95.2 | 86.9 | 94.7 | 95.4 | 95.7 | 94.7 | 94.9 |
| | | QA | 94.5 | 94.5 | 92.8 | 84.0 | 94.1 | 94.2 | 94.1 | 91.0 | 91.1 |
| | | Avg. Rank | 2.7 | 2.3 | 6.0 | 9.0 | 7.0 | 4.3 | 2.3 | 6.7 | 4.7 |
| LLM | Gemma | BBH | 94.7 | 94.7 | 95.1 | 85.9 | 94.5 | 94.1 | 95.0 | 94.3 | 94.5 |
| | | Math | 86.7 | 86.9 | 84.3 | 80.8 | 74.8 | 84.9 | 84.4 | 81.8 | 82.6 |
| | | QA | 91.5 | 91.5 | 90.5 | 88.8 | 89.3 | 90.6 | 90.9 | 89.7 | 89.1 |
| | | Avg. Rank | 2.0 | 2.3 | 3.7 | 8.7 | 7.3 | 5.0 | 3.0 | 6.7 | 6.3 |
| | Granite | BBH | 90.5 | 90.4 | 90.1 | 74.7 | 89.8 | 89.2 | 91.2 | 89.8 | 89.6 |
| | | Math | 89.2 | 89.4 | 87.0 | 85.9 | 73.0 | 87.4 | 87.3 | 84.4 | 84.7 |
| | | QA | 90.3 | 90.4 | 89.0 | 86.8 | 87.3 | 87.8 | 89.9 | 87.0 | 88.3 |
| | | Avg. Rank | 2.0 | 1.7 | 4.3 | 8.0 | 7.3 | 5.7 | 2.7 | 7.0 | 6.3 |
| | Llama | BBH | 91.3 | 91.3 | 90.6 | 84.3 | 91.2 | 90.6 | 91.5 | 89.5 | 90.1 |
| | | Math | 85.6 | 85.6 | 84.4 | 79.0 | 67.8 | 84.0 | 81.5 | 83.3 | 82.1 |
| | | QA | 90.0 | 90.1 | 88.4 | 86.6 | 88.4 | 89.1 | 89.8 | 87.6 | 88.8 |
| | | Avg. Rank | 2.0 | 1.7 | 4.7 | 8.7 | 6.7 | 4.7 | 3.7 | 7.0 | 6.0 |
| | Qwen3 | BBH | 97.7 | 97.7 | 97.9 | 97.5 | 97.3 | 97.8 | 97.9 | 96.9 | 97.1 |
| | | Math | 94.2 | 94.2 | 92.4 | 92.5 | 93.6 | 93.6 | 93.6 | 92.0 | 91.5 |
| | | QA | 91.3 | 91.5 | 89.0 | 89.8 | 89.4 | 90.2 | 90.8 | 87.8 | 87.9 |
| | | Avg. Rank | 2.7 | 2.3 | 5.0 | 5.7 | 5.7 | 3.3 | 3.3 | 8.7 | 8.3 |

Table 2: Comparison in terms of accuracy increase at 10% rejection.

| Setup | Model | Data | ADG | Ens | EDL | GPP | CP | LEns | Prob | GN | GO |
|-------|-------|------|-----|-----|-----|-----|-----|------|------|-----|-----|
| Guardian | GG-3.1 | Harm | 3.6 | 3.5 | 4.0 | 3.6 | 3.0 | 3.3 | 4.1 | 2.9 | 4.0 |
| | | Jail | 3.8 | 3.7 | 3.9 | 3.6 | 0.9 | 3.3 | 3.3 | 2.6 | 4.0 |
| | | True | 3.8 | 3.7 | 2.2 | 1.9 | 0.2 | 2.6 | 3.5 | 2.2 | 2.3 |
| | | Avg. Rank | 2.8 | 4.0 | 3.5 | 5.8 | 8.7 | 5.8 | 3.5 | 8.0 | 2.8 |
| | GG-3.2 | Harm | 4.1 | 3.9 | 3.0 | 3.4 | 3.9 | 3.8 | 3.6 | 2.3 | 3.9 |
| | | Jail | 3.2 | 3.5 | 3.1 | 3.0 | 2.6 | 3.3 | 4.2 | 3.2 | 3.7 |
| | | True | 2.7 | 2.7 | 1.6 | 2.2 | 0.8 | 1.7 | 2.0 | 2.0 | 2.1 |
| | | Avg. Rank | 2.5 | 2.7 | 7.7 | 6.0 | 7.3 | 5.3 | 4.2 | 6.7 | 2.7 |
| Judge | Gemma | BBH | 2.1 | 2.2 | 1.4 | 1.8 | -0.2 | 1.9 | 2.3 | 0.9 | 0.7 |
| | | Math | 1.9 | 1.8 | 2.4 | -0.2 | -1.3 | 1.6 | 2.5 | 1.5 | 1.0 |
| | | QA | 2.6 | 2.5 | 1.8 | 1.7 | -0.0 | 1.8 | 3.6 | 1.5 | 2.1 |
| | | Avg. Rank | 2.7 | 3.0 | 4.5 | 6.7 | 9.0 | 4.8 | 1.0 | 7.0 | 6.3 |
| | Granite | BBH | 2.5 | 2.3 | 3.3 | 1.4 | 0.4 | 2.3 | 2.9 | 2.8 | 2.6 |
| | | Math | 3.8 | 3.6 | 1.6 | 0.7 | -1.5 | 1.7 | 2.5 | 0.7 | 1.5 |
| | | QA | 2.5 | 2.1 | 1.7 | 1.1 | -0.8 | 2.8 | 1.5 | 1.3 | 1.7 |
| | | Avg. Rank | 2.7 | 3.8 | 3.5 | 8.0 | 9.0 | 3.8 | 3.7 | 5.7 | 4.8 |
| | Llama | BBH | 2.1 | 1.7 | 2.2 | 1.8 | 2.1 | 1.7 | 3.4 | 1.2 | 1.2 |
| | | Math | 3.2 | 3.1 | 2.1 | 1.1 | -2.3 | 2.0 | 3.1 | 2.5 | 2.2 |
| | | QA | 2.3 | 2.2 | 0.9 | 1.9 | 0.4 | 2.9 | 2.9 | 2.2 | 2.3 |
| | | Avg. Rank | 2.8 | 4.8 | 5.3 | 6.7 | 7.0 | 5.0 | 1.7 | 6.0 | 5.7 |
| | Qwen3 | BBH | 2.2 | 2.3 | 2.6 | 2.4 | 0.6 | 2.5 | 3.0 | 3.1 | 3.0 |
| | | Math | 2.7 | 2.7 | 3.1 | 0.4 | 0.0 | 2.1 | 3.7 | 2.6 | 2.9 |
| | | QA | 3.0 | 2.9 | 2.0 | 2.0 | 1.5 | 3.1 | 3.0 | 2.4 | 2.3 |
| | | Avg. Rank | 4.8 | 5.3 | 4.5 | 7.2 | 9.0 | 4.3 | 1.8 | 4.0 | 4.0 |
| LLM | Gemma | BBH | 3.3 | 3.2 | 2.8 | 0.4 | 2.3 | 2.2 | 3.4 | 3.0 | 2.6 |
| | | Math | 3.8 | 3.7 | 2.2 | 1.4 | -1.7 | 1.4 | 2.4 | 1.5 | 1.3 |
| | | QA | 3.9 | 3.8 | 2.1 | 2.4 | -0.5 | 2.4 | 3.0 | 2.1 | 1.7 |
| | | Avg. Rank | 1.3 | 2.3 | 5.2 | 6.7 | 8.3 | 6.3 | 2.3 | 5.2 | 7.3 |
| | Granite | BBH | 3.0 | 3.1 | 3.0 | 0.8 | 2.6 | 2.4 | 3.7 | 2.9 | 2.1 |
| | | Math | 4.0 | 4.2 | 2.8 | 2.3 | -1.6 | 1.9 | 3.1 | 1.6 | 1.6 |
| | | QA | 2.0 | 2.3 | 2.2 | 1.6 | -0.6 | 2.3 | 2.6 | 1.7 | 2.2 |
| | | Avg. Rank | 4.0 | 1.8 | 3.8 | 7.3 | 8.0 | 5.2 | 1.7 | 6.5 | 6.7 |
| | Llama | BBH | 2.6 | 2.6 | 2.6 | 2.3 | 2.0 | 1.5 | 3.3 | 2.4 | 2.2 |
| | | Math | 2.9 | 2.9 | 3.0 | 1.2 | -2.2 | 0.4 | 1.6 | 2.2 | 1.6 |
| | | QA | 2.9 | 2.8 | 1.6 | -0.7 | 0.1 | 2.5 | 2.6 | 0.9 | 2.2 |
| | | Avg. Rank | 2.2 | 2.8 | 3.0 | 7.3 | 8.3 | 7.0 | 3.0 | 5.3 | 6.0 |
| | Qwen3 | BBH | 2.4 | 2.3 | 2.8 | 2.5 | 0.4 | 2.9 | 2.9 | 2.0 | 1.7 |
| | | Math | 3.8 | 4.0 | 1.6 | 1.6 | 3.3 | 3.4 | 2.8 | 0.9 | 1.1 |
| | | QA | 3.4 | 3.8 | 2.3 | 0.9 | -0.2 | 2.3 | 2.1 | 1.3 | 1.3 |
| | | Avg. Rank | 3.0 | 2.7 | 4.3 | 6.2 | 7.3 | 2.8 | 3.7 | 7.5 | 7.5 |

