# OpenReview forum: "Uncertainty-Aware LLM Probing"
_ICLR.cc/2026/Conference — Submitted to ICLR 2026_

### Official Review · Reviewer_eQcP · 2025-10-31

**Soundness:** 2
**Presentation:** 2
**Contribution:** 2
**Rating:** 4
**Confidence:** 5

**Summary:**

This paper investigates whether uncertainty quantification methods can improve LLM probe reliability under distribution shifts. The authors evaluate several established UQ techniques and propose two gradient-based scores: JDGrad and ADGrad. Experiments across guardian models and QA benchmarks with multiple LLMs show that most methods perform reasonably in-domain, with deep ensembles and ADGrad ranking best, while JDGrad provides useful OOD detection signals.

**Strengths:**

- The paper identifies a genuine issue in the probing literature, where probes fail under distribution shifts. The motivation for exploring uncertainty quantification methods is reasonable.
- The authors test multiple UQ methods across diverse settings and several LLMs, providing a useful empirical comparison of existing approaches.
- The paper is generally well-structured with clear sections, and the writing is accessible, making it easy to follow.

**Weaknesses:**

I do have some concerns with the paper that I believe should be addressed:

- First, there is a misalignment between the motivation and the contribution of this paper. The paper's title and motivation focus on making probes reliable under distribution shifts. However, I found the main contribution of this paper, JDGrad's OOD detection capability, only identifies when probes might fail, without actually fixing the reliability issue. I believe the core problem the authors should address is making probes work reliably under slight distribution shifts, not just detecting OOD samples after the fact.
- The authors prove in lines 204-207 that $s_{ADGrad}(h) = |1 - 2f_\theta(h)|$, which reveals that ADGrad is simply a transformation of the probe's output probability rather than a new UQ method. More puzzlingly, Fig. 7 shows that "calibrated probe probability" performs best, which should be nearly equivalent to ADGrad. Additionally, the paper does not explain the existence of these discrepancies or offer a theoretical justification for the proposed transformation's benefits.
- I found that an Unfair computational comparison exists. Deep ensembles utilize 50 models, whereas the gradient-based methods are post-hoc computations on a single probe; the paper does not adequately discuss this significant difference in computational cost.
- While the paper claims strong results on guardian models, these tasks have highly constrained characteristics (e.g., fixed prompt templates, binary classification), and Table 1's bottom half shows that improvements on conventional QA tasks are marginal at best. This raises serious concerns about whether the proposed methods generalize beyond the cherry-picked scenarios where they work well.

**Questions:**

Please see the weaknesses I've outlined.

---

> ### Author Response · Authors · 2025-11-24
> **Response to Reviewer eQcP**
>
> We thank the reviewer for the careful and thoughtful assessment of our work.
>
> **Motivation and Contribution Alignment**
>
> - **Topic.**  Our submission does *not* aim to improve or redesign probe architectures, but rather to **reliably detect when existing probes fail**. We view strong failure detection as a necessary first step toward practical probe reliability, which is exactly the goal of uncertainty quantification. We agree that improved probe architectures should be explored in parallel.
>
> - **Title.**  The title aligns with our focus on **uncertainty quantification for LLM probes**. To the best of our knowledge, there is no prior systematic study of UQ methods specifically on LLM probes. While Ovadia et al. (2019) examine UQ under distribution shifts, they do not do so in the probing context. Since our work includes most established UQ approaches, we intentionally kept the title broad.
>
> - **A New Gradient-Based Score.**  Gradient-based scores have recently shown strong OOD detection potential, which motivates our focus on them. Existing methods, such as GradOrth do not reach the best in-domain performance (i.e., deep ensembles perform better), and GradNorm similarly underperforms. We therefore **design a new gradient-based score tailored to binary classification**, which achieves strong in-domain results while retaining OOD detection capability. However, we also show that *small distribution shifts are not reliably detected by any UQ method tested*, which we believe is an important observation in itself and motivates future work. This clarification has been added to the revised manuscript.
>
> **ADGrad Analysis and Theoretical Justification**
>
> We appreciate the opportunity to clarify the theoretical part. The connection to the probability transformation that we prove holds when we set `batch_size = 1`. In this regime, ADGrad reduces to a margin-like quantity. In practical settings with `batch_size > 1`, gradient accumulation breaks this exact equivalence, and ADGrad instead captures **gradient flow patterns across layers**, not just the output probability. We believe this theoretical link is useful for understanding the relationships between scores, the good empirical performance of ADGrad, and its nature as a principled metric.
>
> We also added a simple margin-based UQ score (Prob = |probability − 0.5|) to the new plots to highlight the differences to ADGrad. Interestingly, this simple score itself is among the strongest in-domain baselines, which makes the fact that ADGrad still provides additional value more meaningful.
>
> The application experiment shows that probe UQ scores can have considerable impact compared to standard LLM-based UQ scores in certain settings, even when differences between individual probe UQ methods are smaller. We do not view this as a negative result: in **Figure 2**, we see that several methods clearly underperform (GPP, CP, GN, GO), while deep ensembles and EDL turn out to be overall best among existing approaches. ADGrad/JDGrad emerge as robust competitors alongside these methods.
>
> **Computational Cost / Setup Clarification**
>
> To make the comparison more realistic and consistent with related work, we updated the setting to an ensemble of 10 models (5 architectures × 2 seeds). The overall conclusions did not change.
>
> We also clarified the ensemble computation of all scores. In the revised version, we explicitly state that we average scores across the ensemble to smooth out noise and removed the inconsistently used *total* suffix. In particular, for *all* gradient-based methods (including the baselines), we compute gradients for each model in the ensemble individually, requiring 1 forward + 1 backward pass per label (2 total for binary classification) for each of the 10 models, and then average:
> $
> s_{\text{final}}(h) = \frac{1}{10}\sum_{i=1}^{10} s_{\text{grad}}^{(i)}(h).
> $
> This ensemble-based gradient computation ensures robustness to architectural variation while keeping computational cost comparable to deep ensembles.

---

> ### Author Response · Authors · 2025-11-24
> **Response to Reviewer eQcP Part II**
>
> **Generalization Beyond Guardian Models**
>
> By “focusing on the Guardian”, we mean that we **include** such safety-relevant Guardian and LLM-as-a-Judge settings, rather than ignore them. This stands in contrast to much of the probing literature, which predominantly considers standard benchmarks. We have rephrased the contributions section to avoid misunderstanding.
>
> At the same time, the overall experiments are still dominated by standard benchmarks. As illustrated in the **updated Figure 2 and additional tables in the appendix**, each dot corresponds to one LLM–dataset pair, and there are substantially more such runs for the conventional benchmarks than for the Guardian data.
>
> Overall, we provide *extensive experiments*, beyond what is usual in related work:
> - We evaluate on 8 different LLMs (8B–70B parameters).
> - We test diverse datasets covering factual QA, reasoning, Math, and safety.
> - We added plots for two probe targets beyond correctness (semantic entropy and harm), where the trends mirror those observed for correctness.
>
> Altogether, we observe a consistent pattern:
> - ADGrad and ensembles rank best in-domain and, together with some other methods, provide useful OOD signals.  EDL turns out to be a close competitor.
> - The fundamental issue—failure under subtle shifts—affects *all* methods.
>
> We are happy to provide further clarification or additional analysis if helpful. Thank you again for the constructive feedback.

---

### Official Review · Reviewer_WtiQ · 2025-10-31

**Soundness:** 3
**Presentation:** 2
**Contribution:** 3
**Rating:** 4
**Confidence:** 3

**Summary:**

The authors examine how effective uncertainty quantification methods are at quantifying the uncertainty of LM probes, and design their own gradient-based quantification method. They find that no method is robust to domain shifts, but that their method can detect OOD effectively.

**Strengths:**

1. The method detects OOD data more effectively than other uncertainty quantification methods.
2. Examining the uncertainty of LLM probes is an interesting idea that I have not previously seen explicitly examined.
3. The thoroughness of doing a human analysis of failures demonstrates diligence on the part of the authors, and is a good reminder to take label noise into account, which may be under-discussed in many UQ methods.

**Weaknesses:**

1. The writing overall tends to be unclear and not well-organized. Many sentences lack specificity or need to be better defined. (ex. Line 58 “turns out to be sub-optimal”—how? Lines 75-77, “Based on evaluations on uncertainty quantification in traditional ML, we would expect the uncertainty methods to work reliably here”—which evaluations, and why?)
2. Some of the results would benefit from further analysis, as inconclusive results are reported without investigation into the cause. For instance, lines 313-314, “More precisely, we sometimes observed very high performance, but very low one at other times”, does not present any hypothesis for why this was the case.
3. The performance benefit on ID data does not appear to be consistent and significant. It only outperforms all other methods in “average rank” with 2/5 models. Figure 2 also suggests to me that these metrics all achieve similar scores, which does not present a compelling case for using ADGrad over other established methods.
4. While OOD data detection is useful, it is not immediately obvious why this should be compared to uncertainty quantification methods as baselines and not baselines for OOD data detection. This seems like a separate application.

**Questions:**

Why report the average rank of the scores and not simply average over all categories? This seems somewhat arbitrary.

---

> ### Author Response · Authors · 2025-11-24
> **Response to Reviewer WtiQ**
>
> Thank you for the thorough evaluation. We address each point below.
>
> **Writing**
>
> We apologize for any confusion caused. We have carefully revised the affected parts and highlighted the main updates in blue.
>
> Regarding the examples mentioned:
>
> - **Line 58:** “Azizian et al. (2025) test conformal prediction on LLM probes. The latter turns out to be sub-optimal”—why?
>   We now clarify that conformal prediction relies on an exchangeability assumption, i.e., calibration and test samples must be drawn from the same distribution such that conformity scores remain comparable.
>
> - **Lines 75–77:** “Based on evaluations on uncertainty quantification in traditional ML, we would expect the uncertainty methods to work reliably here”—which evaluations, and why?
>   This refers to the discussion in the preceding paragraph (“Uncertainty-Aware Probes.”), where we cite several works on UQ in conventional ML.
>
> **Human Analysis**
>
> Thank you for recognizing our contributions in this regard.
>
> **Details on Inconclusive Results**
>
> We have revised the results discussion to make it more concrete. Regarding the high variance observed with several baselines, we hypothesize that it stems from the generally lower performance of these methods, which aligns with prior findings. As shown in Figure 2, the better-performing methods exhibit acceptable variance. Contributing factors likely include:
> - dataset-specific label noise,
> - overconfidence of Guardian models on ambiguous samples (seen in Fig. 7),
> - higher proportions of ambiguous samples in test sets (as supported by the case study).
>
> **In-Domain Performance Significance**
>
> To clarify the impact, we have revised and expanded the presentation (**see updated Figure 2**). Stable performance across datasets is key for deployment, and ADGrad and ensembles show the lowest variance across settings. We also included results for two additional probe targets (semantic entropy and harm), for which the trends match those observed for correctness.
>
> **OOD Detection Methodology**
>
> Dual capability (in-domain and OOD) is essential for practical deployment where distribution types are often unknown. Otherwise, uncertainty methods would have to be combined with separate OOD detection.
>
> Related evaluations under distribution shifts have been performed in traditional ML (Ovadia et al., 2019). Here, we examine this question in a modern LLM probing setup. We also evaluate established gradient-based OOD methods (GradNorm, GradOrth), demonstrating that they can be adapted to yield meaningful probe uncertainty scores in-domain.
>
> However, our results show that *all evaluated methods struggle under slight distribution shifts*, which underscores an important limitation to be addressed before applying LLM probes with UQ in practice.
>
> **On the Average Rank Metric**
>
> Ranking based on average performance across experiments can introduce systematic bias due to heterogeneous variance across datasets. In particular, averaging may overweight outlier cases and mask consistent overall trends. A more robust alternative would analyze the full distribution of ranks per dataset or employ rank-based statistical methods that account for variance structure.
>
> In the revised submission, we moved average rank results to the appendix and provide summary plots instead, allowing a clearer view of differences across methods and settings (in-domain vs. shifted).
>
> ---
>
> Thank you again for the valuable feedback. Please let us know if any further clarification is needed.

---

### Official Review · Reviewer_EDim · 2025-11-03

**Soundness:** 3
**Presentation:** 1
**Contribution:** 3
**Rating:** 4
**Confidence:** 3

**Summary:**

Authors study uncertainty quantification of probes, specifically looking at guardian models. They argue that gradient based methods are the only ones that work at all under distribution shift, and improve on them to also be competitive in terms of uncertainty quantification. This improvements is a normalization of the gradients, which avoids variations in scale which could happen in data shifts. Use cases are provided.

**Strengths:**

Litterature review is well explained. Math is quite straightforward, useful, and seems to come for a relevant question. Multiple datasets, baselines, and models are used to empirically validate hypothesis. Extensive results are provided in Appendix to guide future works.

**Weaknesses:**

A reminder of the basics of how Guardian model works would be relevant, they are quite central to the empirical part.

Some dataset choice seems quantitative, all datasets, and the differences between them could be discussed more. Do the fail cases correspond to stronger distributions shifts? are they different to the ones observed by other methods?

The discussion of results was hard to follow for me, require multiple read throughs - perhaps it would help to clarify the setting, and group results in subparagraphs instead of one block.

How usable is the proposed method out of distribution? Is the take home that this new gradient based method is the best OOD, but that it still performs too chaotically to be trusted with critical use cases?

(Minor) There are many unclear sentences, this makes understanding the narrative harder:
e.g.
Line 148/149 "Moreover, give" should probably be "Moreover, we give"?
line 149/150 is BNN for Bayesian Neural Network?
line 157 "get away with the noise" might be "remove some noise"?
line 371 underperfored should be underperformed
etc...

(Minor)
Reproducibility could be improved by providing hyperparameter and prompts.

**Questions:**

see weaknesses

---

> ### Author Response · Authors · 2025-11-24
> **Response to Reviewer EDim**
>
> Thank you for the fair and constructive feedback. We address each point below.
>
> **Guardian Model Clarification**
>
> We have added more explanation in Appendix C. In a nutshell, the Granite Guardian models are regular Granite models (in the case of the 5b model, pruned) that are fine-tuned for binary classification tasks (e.g., harmful content detection, RAG groundedness verification). For each task, there is one fixed prompt template, and the models we consider output a single token, “Yes” (harmful) or “No” (benign); newer versions additionally allow for reasoning. We believe this one-token output format yields an especially clean and suitable case for studying probing. Moreover, we consider uncertainty quantification critical for such safety-relevant models.
>
> **Results Structure / Grouping**
>
> We have restructured the main results discussion accordingly, so that the different regimes and their implications for probe applicability become clearer.
>
> **Performance Out of Distribution**
>
> This feedback was very helpful in clarifying our main findings. We now distinguish three settings more explicitly:
>
> 1. **In-Domain Performance.**
>    ADGrad and ensembles consistently rank best. Other gradient-based scores do not allow for reliable uncertainty quantification, whereas our score performs on par with the currently strongest method, deep ensembles.
>
> 2. **Slight Distribution Shifts.**
>    For instance, choosing train and test data from within the same category (e.g., Harm, Math) but from different datasets. In this setting, performance is unstable for all methods. We **added Figure 2** to emphasize this and view it as a critical finding: since probes already struggle under such shifts, the lack of reliable UQ signals further limits current applicability.
>
> 3. **Out-of-Distribution.**
>    In clear OOD settings, we observe useful signals from multiple scores. Given that ADGrad and ensembles perform strongest in-domain, we focus on comparisons between those and JDGrad. In our experiments, JDGrad performs slightly better. **See updated Figures 2 and 5.**
>
> Overall, the most reliable scores are ADGrad/JDGrad, deep ensembles, and EDL. However, even these methods struggle under slight distribution shifts.
>
> **About Dataset Choices**
>
> Our dataset selection spans both *popular reasoning types* (mathematical reasoning, chain-of-thought, factual QA) and *critical safety use cases* (Guardian settings), enabling evaluation across varying degrees of distribution shift. We grouped datasets into categories that can reasonably represent distinct *distributions*: Math, BBH, and QA for LLM and Judge settings; and Harm, Jail, and True for the Guardian experiments.
>
> **Minor Issues and Reproducibility**
>
> We appreciate the reviewer’s close reading. All minor issues have been addressed.
>
> Complete hyperparameter settings have been added in Appendix C.2 to support reproducibility.
>
> ---
>
> Thank you again, and please let us know if further clarification would be helpful.

---

### Author Response · Authors · 2025-11-24
**Final Summary Comment**

We sincerely thank all reviewers for their constructive and detailed feedback, and for recognizing the contributions of our work.

We revised the manuscript in line with the main reviewer suggestions:

- **Overall writing and clarity** (Reviewers *EDim*, *WtiQ*)
- **Expanded and refined theoretical discussion** (Reviewer *eQcP*)
- **Clarification of experimental settings and computational setup** (Reviewers *EDim*, *eQcP*)
- **Improved summary and interpretation of findings** (Reviewers *EDim*, *WtiQ*, *eQcP*)

In addition, we updated **Figures 2 and 5** to provide a clearer and more concise presentation of the core results.

---

### Meta-Review · Area_Chair_M2LZ · 2026-01-05

**Summary:**

This paper investigates uncertainty quantification methods under distribution shifts. Existing methods are not robust and the paper proposes normalization methods to reduce the variants. Experiments across guardian models and QA benchmarks with multiple LLMs show that it improves OOD detection while maintaining a good in-distribution result. Several reviewer complained the difficulty of fully understanding the paper and the author has improved it during rebuttal.

After reading it, I think one missing piece of the paper is the robustness is not exhaustively evaluated, besides concerns raised by reviewers. On the one side, the paper presented several applications all have alternative baselines (not based on uncertainty) and it is not clear if the method can show improvements over them. On the other hand, the evaluation is not systematic or it is still unclear how the evaluation covers all typical scenarios.

**Reviewer Concerns:**

Reviewer EDim asked a lot of questions about details, from technical details to evaluation settings, echoing similar concerns with Reviewer WtiQ expressed on writing (lacking background, techniques) and missing more analysis of the evaluation, etc. Reviewer eQcP also questioned about the evaluation choices. I believe that the evaluation concerns are still outstanding. The settings have been explained but raised more questions to me.

**Reviewer Scores:**

All reviewers rated the paper with 4, slightly below the bar. I think part of the concerns have been addressed. I will not be surprised if the authors can get higher ratings by clarifying more details. However, the reviewers concerns are diverse and I think more questions will also be raised, especially considering that not only one reviewer could not fully understand the paper originally.

---

### Decision · Program_Chairs · 2026-01-26

Reject